# Removal of Pb(II) Ions Using Polymer Inclusion Membranes Containing Calix[4]resorcinarene Derivative as Ion Carrier

**DOI:** 10.3390/polym11122111

**Published:** 2019-12-16

**Authors:** Iwona Zawierucha, Anna Nowik-Zajac, Cezary A. Kozlowski

**Affiliations:** Institute of Chemistry, Jan Dlugosz University in Czestochowa, Armii Krajowej 13/15, 42-200 Czestochowa, Poland; a.zajac@ajd.czest.pl (A.N.-Z.); c.kozlowski@ajd.czest.pl (C.A.K.)

**Keywords:** calix[4]resorcinarene, polymer inclusion membranes, Pb(II) ions

## Abstract

Stricter environmental regulations regarding the discharge of toxic metals require developing various technologies for the removal of these metals from polluted effluents. The removal of toxic metal ions using immobilized membranes with doped ligands is a promising approach for enhancing environmental quality, because of the high selectivity and removal efficiency, high stability, and low energy requirements of the membranes. Cellulose triacetate-based polymer inclusion membranes (PIMs), with calix[4]resorcinarene derivative as an ion carrier, were analyzed to determine their ability for removal of Pb(II) ions from aqueous solutions. The effects of ion carrier concentration, plasticizer amount, pH of source aqueous phase, and receiving agents on the effective transport of Pb(II) were determined. All studied parameters were found to be important factors for the transport of Pb(II) ions. The PIM containing calix[4]resorcinarene derivative as an ion carrier showed high stability and excellent transport activity for selective removal of Pb(II) from the battery industry effluent, with a separation efficiency of 90%.

## 1. Introduction

The presence of heavy metals in the environment is very harmful because of their toxicity. It is thus very important to mitigate the environmental risk posed by these heavy metals. The contamination of water bodies by toxic metal ions has steadily increased over the last years because of overpopulation, improper hazardous waste storage, and industrial activities [1].

Lead (Pb(II)) is a toxic metal that is mainly introduced into the environment through various industrial applications such as batteries, fuels, paints, constructions, and foundries. Because of its toxicity and nonbiodegradable nature, Pb(II) contamination has attracted attention worldwide [2,3]. Pb(II) adversely affects various body systems, including the neurologic, hematologic, gastrointestinal, cardiovascular, and renal systems. Moreover, Pb(II) causes severe nervous system disorders in young children and damages the blood–brain barrier. The U.S. Environmental Protection Agency (USEPA) and WHO have set the permissible limit of Pb(II) in drinking water as 10 μg/L [4].

Therefore, the removal of toxic metals from wastewater is important to protect public health. To date, various techniques including chemical precipitation, extraction, ion exchange, membrane separation, and sorption, have been developed for effectively removal of heavy metals at various concentrations from different aqueous solutions [5,6,7,8].

The polymer inclusion membrane (PIM) separation can be an attractive option to remove metal ions from aqueous waste streams with very high efficiency. These liquid membrane systems are driven by extraction and re-extraction processes, which occur simultaneously in both interfaces of membranes. To enable ion transport (ion-exchange mechanism), the PIM consists of a polymer (e.g., cellulose triacetate (CTA)), a plasticizer (e.g., *o*-nitrophenyl alkyl ether), and an ion carrier. The resulting membrane is used to separate the source and receiving aqueous phases [9,10]. 

High selectiveness and improved stability are the greatest advantages of polymer inclusion membranes. The immobilization of the ion carrier in the solid polymer matrix guarantees membrane flexibility, while the inclusion of the plasticizer significantly promotes the penetration of metal ions [6]. Moreover, PIMs have mechanical properties that are quite similar to those of filtration membranes, thus enabling PIM-based systems to offer many advantages, such as ease of operation, minimum use of hazardous chemicals, and variability in membrane composition, to achieve the desired selectivity and separation efficiency [11,12]. 

In PIMs, transport of ions resulted from acting of a carrier that works as a complexing factor or an ion exchanger. The complex or ion pair formed in the membrane between the metal ion and the carrier is stable and enables the transfer of metal ions across the membrane. Various types of carriers display diverse transport efficiencies owing to differences in their pathways of complexation. The most significant factors affecting membrane selectivity are the carrier’s chemical structure and its activity in the processes of complexation and transport. The molecular moiety of the carrier can be adapted to achieve a specific selectivity [13].

Calixarenes and their derivatives are a new group of highly efficient carriers for heavy metals and transition ions in PIM separation technology. This effectivity probably results from their high complexing selectivity for metal ions, owing to specifically tailored encapsulating coordination sites present in their structures and their low solubility in aqueous solutions [14]. Researchers have developed a high interest in these macrocyclic ligands because of their simplicity of synthesis, high yield, and ability to act with different functional groups in their aromatic core (narrow or broad rim) [15].

Modified calixarenes have acceptable complexation–decomplexation dynamics, together with the likelihood of three-dimensional cation encapsulation by the calixarene unit and the sidearm. To effectively transfer metal ions in a separation process, a counter anion that is soluble in both aqueous and organic phases should be included in the system. Nevertheless, the diffusion coefficients between an aqueous phase and an organic phase of complexes formed with specific anions such as chlorides, sulfates, or nitrates are too low to be useful. A calixarene unit with an attached proton-ionizable sidearm can reduce the necessity of moving of aqueous phase anions into the organic phase. Another benefit of the use of proton-ionizable calixarenes as ion carriers in liquid membranes’ transport processes is that they enable coupling of metal ion transfer from the aqueous source phase into the aqueous receiving phase with back transport of a proton. Thus, a pH gradient creates a pathway for the transport of metal ion. Sgarlata et al. have synthesized and studied various proton-ionizable calixarenes [16]. Hui-Min et al. [17] investigated the selective transport of Cu^2+^, Fe^3+^, Co^2+^, Ni^2+^, and Zn^2+^ through liquid membranes with new calix[4]crowns. One of those calixcrowns, 25,27-dihydroxy-26,28-(3’,6’-dioxa-2’,7’-dioxooctylene)dioxy-calix[4]arene, showed efficient transport of Cu^2+^ cations. Alpoguz et al. [18] analyzed the cotransport of metal ions (Hg^2+^, Pb^2+^, and Na^+^) across a bulk liquid membrane with calix[4]arene nitrile derivatives as ion carriers. On the basis of the transport rates, both nitrile derivatives were found to be efficient and selective for Hg(II) ions compared with Na+ and Pb2+ ions, and the dinitrile derivative was a better agent than the tetranitrile derivative. Ulewicz et al. assessed competitive transport of a mixture of Zn(II), Cd(II), and Pb(II) cations using PIMs with calix[4]crown-6 derivatives p-tert-butylcalix[4]arene derivatives, as ion carriers [19]. The maximum percentage of Pb(II) removal increased in the following order of R groups attached to the compounds: –OH < –OCH_3_ < –OCH_2_COOH < –OCH_2_COOC_2_H_5_ < –OCH_2_CONHOCH_2_C_5_H_6_. The efficient transport of Cd(II) and Pb(II) ions across bulk and supported liquid membranes was noted using thiacalix[4]arene and p-t-butylthiacalix[4]arene [20,21].

The present study analyzed the removal of Pb(II) ions using the plasticizer membrane technique, with calixarene derivative as the ion carrier. Pb(II) transport efficiency through PIM was assessed by examining the effects of carrier concentration, plasticizer amount in the membrane, source phase acidity, type of receiving phase, and membrane stability.

## 2. Materials and Methods 

### 2.1. Reagents

Inorganic chemicals, Pb(NO_3_)_2_, acetic acid, HCl, and nitric acid, were of analytical grade and purchased from POCh (Gliwice, Poland). Organic reagents, CTA, *o*-nitrophenyl octyl ether (*o*-NPOE), and tetrahydrofuran, were also of analytical grade and purchased from Fluka (Seelze, Germany); they were used as received. Aqueous solutions were prepared with deionized water, with a conductivity of 0.1 µS·cm^−1^.

### 2.2. Synthesis

As described earlier [22], calix[4]resorcinarene compound (Figure 1) was synthesized in sufficient yield by condensation reactions.

The structure of the synthesized derivative of calix[4]resorcinarene was confirmed by ^1^H NMR spectroscopy (Bruker AVANCE 200, Billerica, MA, USA) (500 MHz, DMSO-d_6_), δ: 5.63 (s, 4H, ArH), 6.13 (s, 4H, ArH), 6.73 (d,8H, J = 7 Hz, Ph), 6.96 (m, 16H, Ph, Ar–CH–Ar), 8.56 (s, 8H, ArOH). 

### 2.3. Preparation of PIMs 

A solution in tetrahydrofuran as the organic solvent was prepared by combining CTA as the support, o NPOE as the plasticizer, and calix[4]resorcinarene as the ion carrier (Figure 1). A specified portion of this organic solution was poured into a glass Petri dish with a 5.0 cm glass ring attached to a glass plate using CTA–tetrahydrofuran as a glue. Tetrahydrofuran was evaporated overnight, and the resulting membrane was separated from the glass plate by immersing in cold water. The average CTA membrane thickness was 25 μm (measured by A2002M type digital ultrameter from Inco-Veritas), with 1.0 μm standard deviation over four readings. The effective surface area of the membrane was 5.0 cm^2^.

### 2.4. Transport Studies 

The experimental transport studies were carried out in a permeation cell described by Kozlowski [23], in which the membrane sheet was closely locked between two chambers. The aqueous source phase was 0.0010 M Pb(NO_3_)_2_ (50 cm^3^) and the aqueous receiving phases were distilled water, 0.10 M nitric acid, HCl, or acetic acid (50 cm^3^). Transport was performed at room temperature (20 °C), and the solutions (source and receiving phases) were agitated at 600 rpm using synchronous stirrers. The portions of the source and receiving samples were taken regularly through a sampling channel using a pipette, and analyzed to determine Pb(II) concentration. The pH of source phase was checked by a pH meter (multifunctional pH meter, CX-731 Elmetron, with combined pH electrode, ERH-136, Hydromet, Poland). 

The rate of change of concentration of Pb(II) ions in the source, receiving phase, and membrane was analyzed for the transport of Pb(II) ions through PIM containing calix[4]resorcinarene, providing concentration profile of the metal as a function of time (Figure 2). As seen in Figure 2, in the case of the membrane, there is no accumulation of metal, which confirms that the rate of Pb(II) transfer to and from the membrane was comparable and fast.

The kinetics of transport process across PIM is defined by the first-order reaction with regard to the metal ion concentration [24]:(1)$$\ln c/c_{i}=-\mathrm{kt},$$
where c is the metal ion concentration (mol·dm^−3^) in the source phase at a given time, c_i_ is the initial Pb(II) concentration in the source phase, k is the rate constant (s^−1^), and t is the transport time (s).

To determine the k value, a figure of ln(c/c_i_) versus time was drawn. The rate constant value was calculated, and the permeability coefficient (P) was given from Equation (2), as follows:(2)P=−(V/A)k,
where V is the volume of aqueous source phase and A is the area of membrane.

The initial flux (J_i_) was defined as follows:(3)$$J_{i}={\mathrm{Pc}}_{i}.$$

The transport efficiency (recovery factor—RF), that is, metal ion removal from the source phase, was determined as follows:(4)$$\mathrm{RF}=\frac{c_{i}-c}{c_{i}}\cdot 100\%.$$

The lead(II)concentrations were analyzed by flame atomic absorption spectrometry (Solar 939, Unicam, Thermo Fisher Scientific, Waltham, MA, USA). 

The reported values correlated with the average of three replicates, and the observed standard deviation was less than 2%.

## 3. Results and Discussion

The most important factors affecting the transport of metal ions through PIMs are the acidity of the source phase, plasticizer amount, carrier concentration in the membrane, and type of receiving phase. 

### 3.1. Effect of Acidity of the Source Phase

The first series of experiments assessed the effect of pH of the aqueous source phase on the transport of Pb(II) ions from aqueous nitrate solutions containing the metal ion species at 0.0010 M concentration. The acidity of source phases varied from 1.0 to 6.0. The membrane phase containing 0.30 M calix[4]resorcinarene based on plasticizer volume and 0.1 M HCl as the receiving phase were used in the transport process. As shown in Figure 3, an increase in pH of the aqueous solution correspondingly increased the fluxes of the metal ion. The increase in the extraction is the result of the effect of hydrogen and nitrate ions, which enhances the formation of extractable metal complex species in the membrane phase (occurs as a counter-ion). The metal carrier complex is transported through the membrane from the source to the receiving phase, and the counter-ion is transported in the opposite direction. It was found that Pb(II) ion transport increases with the increase in the pH of the aqueous source phase up to 5.0. Therefore, a pH gradient provides the path for transport of metal ion.

Transport of metal ions using calix[4]resorcinarenes depends on the facilitated counter transport [14]. In our study, the source phase and the receiving phase showed a significant difference in pH. These results suggest that metal ion transport occurs where the driving force is generated by the complexation of metal ions at the source phase/membrane and the membrane/receiving phase and by the difference in proton concentrations in both aqueous solutions.

There are probably two ways of transport of metal ions through PIMs with calix[4]resorcinarenes: (i) host–guest adhesion of metal ions in the cavity of ligand [25] and (ii) interaction of metal ions with the OH group rim and by an electrostatic interaction between aromatic rings and the metal ion, as well as by the cation–π interaction [26]. The present study found that the factors that influenced the transport of Pb(II) ions using PIM with calix[4]resorcinarene as an ion carrier were the electrostatic interaction with the OH group rim and the cation–π interaction.

### 3.2. Effect of the Type of Agents in the Receiving Phase

The efficiency of metal ion transport could be significantly influenced by the nature and composition of the receiving phase. Table 1 lists the percentages and fluxes of Pb(II) ion transport with different receiving agents under similar experimental conditions.

As shown in Table 1, the best efficiency of Pb(II) ion transport was achieved when 0.1 M HCl was used as the receiving phase as compared with the other receiving agents. The total percentage of lead ions in the source and receiving phases does not equal 100; this is because some Pb(II) ions remain in the membrane phase, especially when the stripping agent does not possess a high enough tendency to completely release Pb ions from the Pb(II)-calix[4]resorcinarene-NO_3_- adducts (included in the membrane phase) into the receiving phase.

### 3.3. Effect of Carrier Concentration

Pb(II) ion transport through PIMs with various concentrations of calix[4]resorcinarene derivative as the ion carrier was investigated. Blank experiments with no carrier showed no significant flux across PIMs containing only the support and plasticizer. The effect of carrier concentration on Pb(II) ion transport at the carrier concentration range of 0.025–0.5 M (based on volume plasticizer) was investigated. The results are shown in Figure 4.

It was found that with the increase of carrier content in membrane (up to 0.3 M) the transport of Pb(II) ions was more effective. The Pb(II) flux was then controlled by the carrier’s accumulation on the membrane surface. With a higher carrier content, the complexation rate of Pb(II) and calix[4]resorcinarene at the interface of PIM was promoted, which led to higher concentration gradient of the complex in the membrane. Accordingly, the transport driving force became greater. However, when the ligand concentration was over 0.3 M, the density of the membrane also increased, which reduced the diffusion of Pb(II)-calix[4]resorcinarene complex in the membrane. The transport flux then tended to slightly decrease. Therefore, the optimal carrier concentration was maintained as 0.3 M. As shown in Figure 5, the carrier concentration of 0.3 M also yielded the best results for the recovery factor (RF).

The maximum flux value (9.7 μmol/m^2^s) and the highest RF value (94%) were obtained for PIM with a 0.3 M carrier concentration. All other flux and RF values were lower for other carrier concentrations, which can be explained by the saturation effect of the Pb(II)–carrier complex in the membrane. 

The membrane matrix with CTA is a polar polymer with several acetyl groups that are capable of forming highly orientated hydrogen bonds. The polarity and crystalline nature of the CTA polymer may lead to incompatibility with macrocyclic carriers at their higher concentrations. In this case, the crystalline precipitates of macrocyclic carrier tend to be formed on the surface of the membrane, preventing the formation of membrane pores.

Lead is a well-known toxic metal that can be separated from other metals using immobilized membrane separation processes. Different macrocyclic carriers were used to extract Pb(II) with various PIMs (Table 2).

As shown in Table 2, the obtained flux (9.7 μmol/m^2^s) was significantly higher for the newly developed membrane with calix[4]resorcinarene than for other membranes reported in the literature. 

### 3.4. Effect of Plasticizer Amount in the Membrane

To investigate how the plasticizer amount affects lead transport efficiency, membranes were prepared using a constant carrier concentration of 0.3 M and different amounts of o-NPOE/1.0 g CTA (1.0–3.0 cm^3^). Figure 6 shows the relationship between fluxes and the amount of o-NPOE. The transport fluxes increased with the increase in the amount of plasticizer until a specific value (2.0 cm^3^ o-NPOE/1.0 g CTA) was reached. A further increase in the plasticizer amount decreased the permeability of the membrane. An increase in the amount of *o*-NPOE, which has a high dielectric constant, subsequently increased membrane thickness and viscosity, which begins to decrease after reaching its maximum value.

The image (3D scan) obtained by atomic force microscopy (AFM) for membrane contains an optimum amount of ion carrier and the plasticizer is shown in Figure 7. The pore size on organic inclusion (was estimated by visualization method) is clearly visible as small, well-defined dark areas. The middle-sized pores estimated for optimal PIM (29% CTA, 58% plasticizer, 13% carrier) were equal to 0.06 µm.

CTA membranes have porous structures, and thus the distribution of pores is almost uniform (porosity 50%) [24]. A plasticizer and a carrier cover the pores in a CTA matrix. The carrier crystallizes within the membrane and the surface texture is relatively homogeneous, with various porosities and roughnesses [29].

### 3.5. Membrane Stability and Reusability

Among the different properties of the PIM membrane, their strength and stability afford biggest advantages over other liquid membranes, which ensures their applicability on the industrial scale. As shown in Figure 8, the initial flux (and thus the membrane permeability) is constant during 15 cycles (6 h each) of transport experiments using the optimal PIM. These results show that the membrane with the calix[4]resorcinarene derivative has very good transport properties and that there are no significant changes in its properties. All 15 experiments showed efficient transport of Pb ions into the receiving phase, and Pb(II) recovery was then ≥90%.

### 3.6. Effect of Temperature on the Transport Process

The effect of temperature on the transport of Pb(II) ions was studied in the temperature range of 288–313 K. Temperature was varied using a circulating water bath. Table 3 shows the experimental results. 

As shown in Table 3, the values of kinetic parameters correspondingly increase as the temperature increases. Optimum results for the transport process were achieved at 313 K. The activation energy of the transport process was calculated at various temperatures according to Equation (5):(5)$$\ln k=\ln A-E_{a}/A\cdot \left(1/T\right),$$
where k is the reaction rate constant, E_a_ is the activation energy, A is the pre-exponential factor, T is the absolute temperature, and R is the gas constant. 

Figure 9 shows the plot of ln k values against 1/T for Pb(II) ion transport across PIM as a function of temperature. 

As known, the temperature strongly influences on diffusion rate constants. The energy activation values for diffusion-controlled mechanisms are rather small; these values are below 20 kJ/mol [30]. In our study, the activation energy value was found to be 19.1 kJ/mol. This indicates that the transport of Pb(II) ions across PIM with the calix[4]resorcinarene derivative is a diffusion-controlled process.

### 3.7. Selective Removal of Pb(II) from the Battery Industry Effluent 

The optimal PIM was examined for its application to extract lead from the battery industry wastewater. The effluent was acidic (pH = 4.5) with Pb(II) concentration of 194 mg/dm^3^ and presence of other heavy metals like Cd, Zn, and Cu at concentrations of 137, 24, and 11 mg/L, respectively. The Pb(II) transport studies were examined using PIM of composition of 29% CTA, 58% plasticizer, and 13% carrier (weight), with a source of industrial effluent, and 0.1 M HCl as the receiving phase. The obtained transport flux of Pb(II) ions was 8.3 µmol/m^2^s. The removal efficiency of lead from wastewater was 90% after 6 h of continuous operating time (Figure 10). In addition, the RF values for other metals were below 2%. Therefore, the prepared calix[4]resorcinarene-based PIM can be used for the selective removal of lead from industrial wastewater containing a mixture of metal ions.

## 4. Conclusions

The technique of plasticizer membranes with calix[4]resorcinarene as ion carrier was used to achieve the facilitated transport of Pb(II) ions. The parameters (i.e., carrier concentration, plasticizer amount in the membrane, acidity of the source phase, and type of the receiving phase) affecting the transport of lead across PIM were investigated to obtain the best operating conditions for maximum removal of lead from aqueous solutions. Under optimized conditions (i.e., source phase: 1.0∙10−3 M Pb(NO3)2 of pH = 5; membrane: 29% CTA, 58% plasticizer, 13% carrier (weight); receiving phase: 0.1 M HCl), the transport efficiency of Pb(II) ions was 94% after 6 h. The obtained flux (9.7 μmol/m2s) for transport of Pb(II) ions through the membrane was much higher than those reported in other studies using PIMs. An interesting finding related to the newly developed membrane with calix[4]resorcinarene is the mechanism of Pb(II) ions transport by a diffusion-controlled process (Ea = 19.1 kJ/mol). 

The newly developed PIM containing calix[4]resorcinarene derivative as ion carrier was stable and highly permeable. The results of the present study indicate that the transport efficiency of PIM can be reproduced and that a PIM is useful for long-term transport process.

Because of their excellent properties, PIMs with immobilized calix[4]resorcinarenes as carriers might be a promising approach for future studies on the separation of lead from other metals and for applying these plasticized membranes on a larger industrial scale.

## Figures and Tables

**Figure 1 polymers-11-02111-f001:**
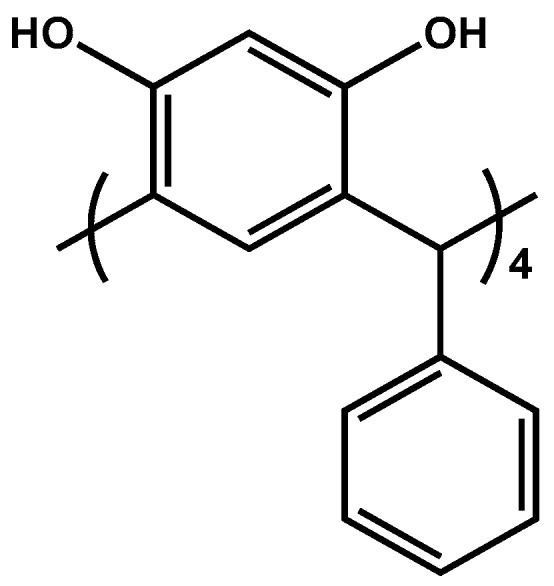
Structure of calix[4]resorcinarene.

**Figure 2 polymers-11-02111-f002:**
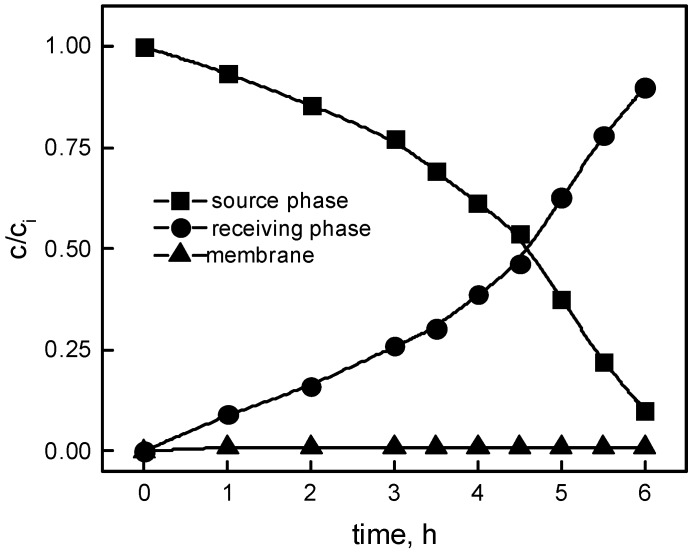
The profile of Pb(II) concentrations in the source, membrane, and receiving phases during the transport process across the polymer inclusion membrane (PIM). Source phase: 1.0∙10^−3^ M Pb(NO_3_)_2_ (pH = 4); membrane: 2.0 cm^3^
*o*-nitrophenyl octyl ether (*o*-NPOE)/1.0 g cellulose triacetate (CTA); 0.2 M carrier; receiving phase: 0.1 M HCl.

**Figure 3 polymers-11-02111-f003:**
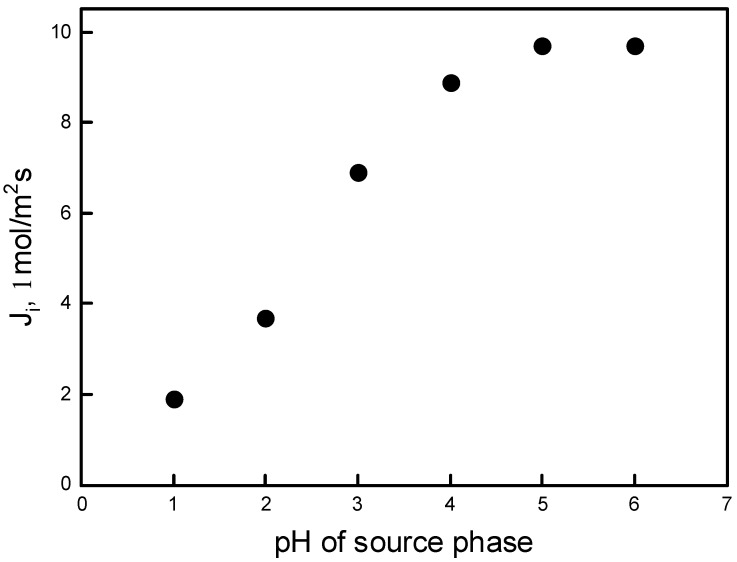
Relationship of the initial ion flux transport Pb(II) across PIM vs. the pH of the source phase. Source phase: 1.0∙10^−3^ M Pb(NO_3_)_2_; membrane: 2.0 cm^3^
*o*-NPOE/1.0 g CTA; 0.3 M carrier; receiving phase: 0.1 M HCl.

**Figure 4 polymers-11-02111-f004:**
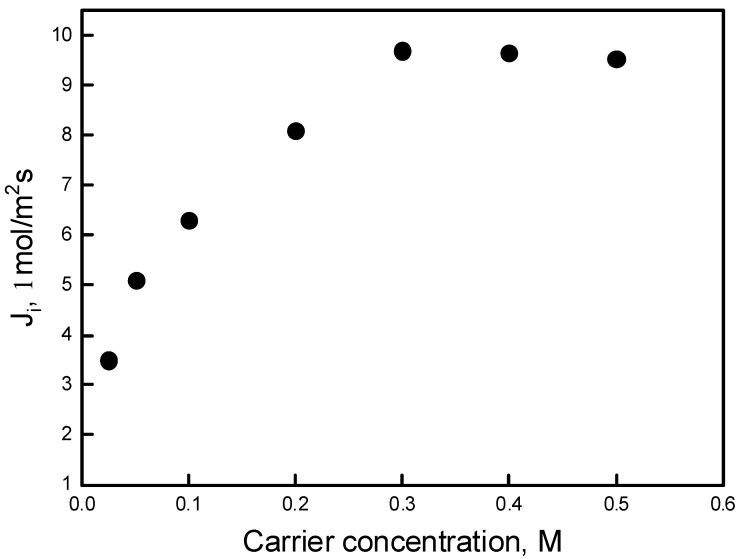
Pb(II) transport fluxes vs. ion carrier concentration by PIM. Source phase: 1.0∙10^−3^ M Pb(NO_3_)_2_, pH = 5.0; membrane: 2.0 cm^3^
*o*-NPOE/1.0 g CTA; receiving phase: 0.1 M HCl.

**Figure 5 polymers-11-02111-f005:**
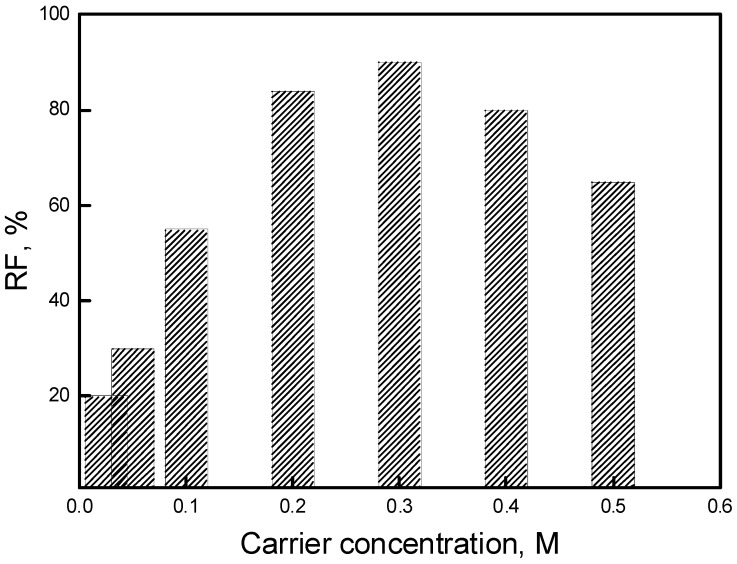
Relationship of recovery factor (RF) of Pb(II) ions vs. carrier concentration. Source phase: 1.0∙10^−3^ M Pb(NO_3_)_2_, pH = 5.0; membrane: 2.0 cm^3^
*o*-NPOE/1.0 g CTA; receiving phase: 0.1 M HCl.

**Figure 6 polymers-11-02111-f006:**
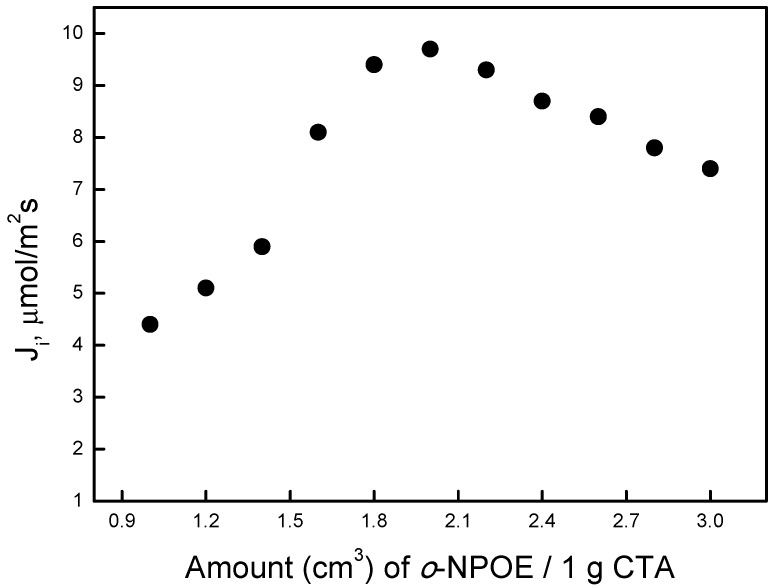
Change in transport fluxes with the amount of o-NPOE. Source phase: 1.0∙10^−3^M Pb(NO_3_)_2_, pH = 5.0; membrane: 2.0 cm^3^
*o*-NPOE/1.0 g CTA, 0.3 M carrier; receiving phase: 0.1 M HCl.

**Figure 7 polymers-11-02111-f007:**
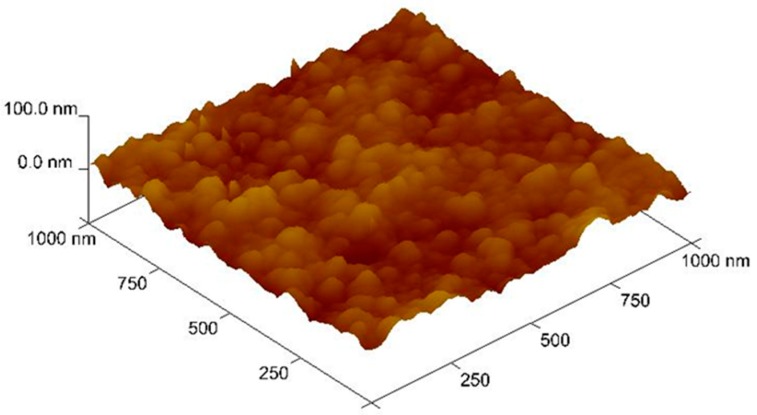
Three-dimensional image of atomic force microscopy (AFM) for polymer inclusion membrane containing 0.3 M carrier and 2.0 cm^3^
*o*-NPOE/1.0 g CTA.

**Figure 8 polymers-11-02111-f008:**
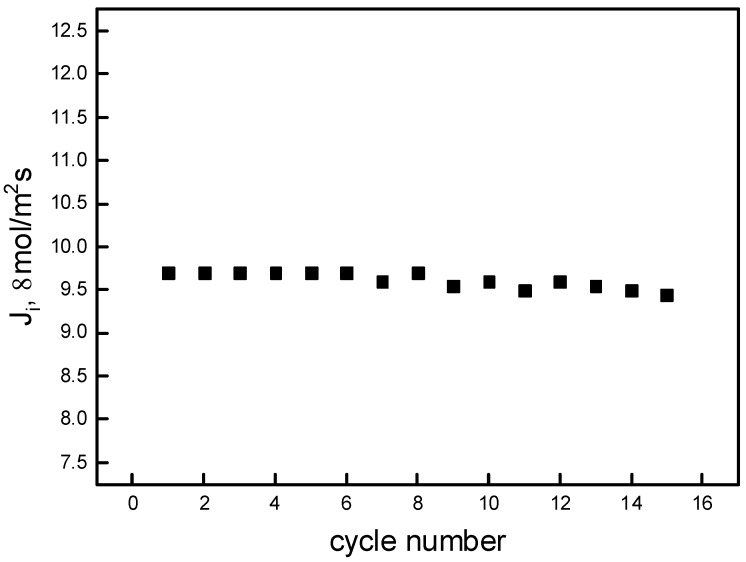
Flux values in consecutive transport experiments performed on the same membrane. Source phase: 1.0∙10^−3^ M Pb(NO_3_)_2_, pH = 5.0; membrane: 2.0 cm^3^
*o*-NPOE/1.0 g CTA, 0.3 M carrier; receiving phase: 0.1 M HCl.

**Figure 9 polymers-11-02111-f009:**
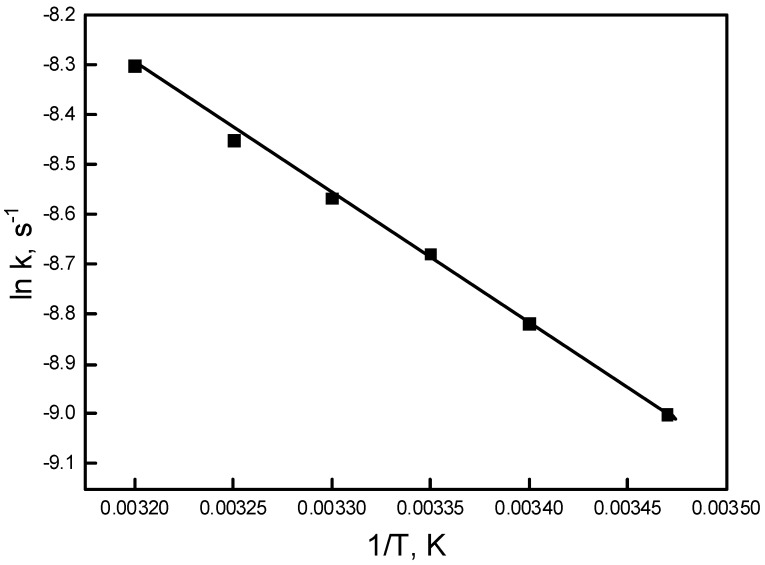
Arrhenius plot of Pb(II) ion transport across PIM with calix[4]resorcinarene. Source phase: 1.0∙10−3 M Pb(NO3)2, pH = 5.0; membrane: 2.0 cm3 o-NPOE/1.0 g CTA; 0.3 M carrier, receiving phase: 0.1 M HCl.

**Figure 10 polymers-11-02111-f010:**
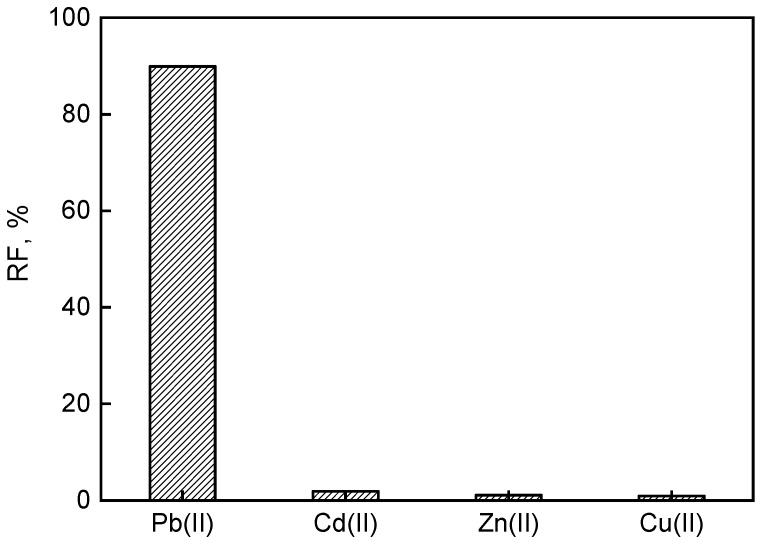
The RF values obtained in competitive PIM transport of metal ions from the battery industrial effluent. Membrane: 2.0 cm^3^
*o*-NPOE/1.0 g CTA; 0.3 M carrier, receiving phase: 0.1 M HCl.

**Table 1 polymers-11-02111-t001:** Effect of the nature of the receiving phase on the transport of Pb(II) ions across polymer inclusion membrane (PIMs). Source phase: 1.0∙10^−3^ M Pb(NO_3_)_2_; membrane: 2.0 cm^3^
*o*-nitrophenyl octyl ether (*o*-NPOE)/1.0 g cellulose triacetate (CTA); 0.3 M carrier.

Receiving agent	Percentage of Pb(II) transported into the receiving phase	Percentage of Pb(II) remaining in the source phase	Flux (µmol/m^2^s)
0.1 M HCl	94	5	9.7
0.1 M HNO_3_	65	10	5.6
0.1 M CH_3_COOH	44	20	4.2
Distilled water	5	9	1.8

**Table 2 polymers-11-02111-t002:** Comparison of Pb(II) transport parameters reported in the literature for plasticized PIMs and those obtained in this work (c Pb(II) = 1.0·10^−3^ M).

Membrane	Membrane composition	Membrane thickness (µm)	Pb(II) initial flux J·10^6^ (mol·s^−1^·m^−2^)	Ref.
PIM	CTA/azacrown ethers/o-nitrophenyl pentyl ether (o-NPPE)	30–35	1.79	[27]
PIM	CTA/PNP-16-crown-6 derivatives/o-NPPE	25	2.24	[28]
PIM	CTA/calix[4]crown-6 derivatives/o-NPPE	35	5.56	[19]
PIM	CTA/β-CD polymer/o-NPPE	28	1.25	[6]
PIM	CTA/calix[4]resorcinarene/o-NPOE	25	9.7	This study

**Table 3 polymers-11-02111-t003:** Effect of temperature on Pb(II) transport across PIM. Source phase: 1.0∙10^−3^ M Pb(NO_3_)_2_; membrane: 2.0 cm^3^
*o*-NPOE/1.0 g CTA; 0.3 M carrier; receiving phase: 0.1 M HCl. RF, recovery factor.

Temperature (K)	k·10^4^ (s^−1^)	P·10^6^ (m/s)	J·10^6^ (mol/m^2^s)	D·10^9^ (m^2^/s)	RF (%)	Transport time (h)
288	3.49	9.45	8.5	4.9	91	6
293	3.71	10.63	9.7	5.1	94	6
298	3.84	11.23	11.9	5.6	96	6
303	4.09	12.46	13.6	6.1	97	5
308	4.44	14.67	14.2	6.5	98	5
313	5.8	18.03	16.4	7.2	99	5

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
