# Peer review of "Removal of Pb(II) Ions Using Polymer Inclusion Membranes Containing Calix[4]resorcinarene Derivative as Ion Carrier"

_polymers, 2019, doi:10.3390/polym11122111_

Round 1

Reviewer 1 Report

The paper "Removal of Pb(II) ions using polymer inclusion membranes containing calix[4]resorcinarene derivative as ion carrier", authored by Zawierucha, I., Nowik-Zajac, A., Kozlowski, C.A.,  describes the potential of cellulose triacetate-based polymer inclusion membranes, with calix[4]resorcinarene derivative as ion carrier, to be used efficiently to be used to remove Pb(II) ions.

Overall, the paper is well written and covers all requested aspects for the interest subject. However, I have the feeling that the paper lacks of novelty since it is mainly related to a standard approach for the characterization of a polymer inclusion membrane (PIM). 

The paper might rise some interest mainly related the investigated PIMs selectivity under optimal conditions. 

The authors must check potential existent typesetting problems (very few).

In giving the information in Table 1 the authors should take into account a possible criteria such as to present the data in a succession easier to be followed.

Moreover, the authors mention that the membrane is stable during 15 cycles, an assertion which to my opinion should be reviewed. Data in Figure 7 show clear evidences about membrane stability during the first eight cycles, while all other tests up to the 15th cycle reveal a lost in the investigated membrane efficiency.

The paper should however come into the attention of researchers dealing with the subject.

I recommend publication after revising the very few minor typesetting problems. Attention should be given by the authors also to the way literature is given in the text.

Reviewer 2 Report

Manuscript tittle: Removal of Pb(II) ions using polymer inclusion membranes containing calix[4]resorcinarene derivative as ion carrier

The manuscript presents an interesting approach for Pb(II) recovery using polymer inclusion membranes. In my opinion, it deserves its publication with some modifications.

The following items should be clarified:

-The use of THF together with CTA is not common. Could you include a reference?

-The composition of the PIMs is not clear. The concentration of the three components in % (weight) should be presented.

-The transport cell dimensions are not given. Only the volume of the receiving phase (50 cm3) is indicated.

-The number of replicates is not indicated.

-In this kind of work dealing with the transport of species, a plot showing the concentration profile vs. time needs to be included.

-In section 3.1, the type of mechanism responsible for the extraction of Pb(II) is not clear. What does it mean “The increase in the extraction is due to the effect of hydrogen and chloride ions”? Is there some chloride ion in the source solution?

-Define NPPE in table 2.

-The authors describe the presence of pores in the PIM. Usually, PIMs are considered non-porous membranes. Please add some references where this particular item is discussed.

-In Figure 5, the membrane thickness would probably change depending on the plasticizer content. Do you think that this has some influence in the variation of the initial flux?

-In section 3.6, the activation energy for the process is calculated and from that value it is said that for the transport of Pb(II) is a diffusion-controlled process for this particular PIM containing the calix[4]resorcinarene derivative as a carrier. If this is true, then the agitation rate would have no influence in the transport. Do they check this? Is there any other evidence for diffusion being the limiting step?

-Finally, subscripts and superscripts should be checked in the text.
